# Structural Knowledge Distillation for Object Detection

Philip de Rijk[1,2]    Lukas Schneider[2]    Marius Cordts[2]    Dariu M. Gavrila[1]

[1]TU Delft    [2]Mercedes-Benz AG

## Abstract

Knowledge Distillation (KD) is a well-known training paradigm in deep neural networks where knowledge acquired by a large teacher model is transferred to a small student. KD has proven to be an effective technique to significantly improve the student's performance for various tasks including object detection. As such, KD techniques mostly rely on guidance at the intermediate feature level, which is typically implemented by minimizing an $\ell_p$-norm distance between teacher and student activations during training. In this paper, we propose a replacement for the pixel-wise independent $\ell_p$-norm based on the structural similarity (SSIM) [28]. By taking into account additional contrast and structural cues, feature importance, correlation and spatial dependence in the feature space are considered in the loss formulation. Extensive experiments on MSCOCO [16] demonstrate the effectiveness of our method across different training schemes and architectures. Our method adds only little computational overhead, is straightforward to implement and at the same time it significantly outperforms the standard $\ell_p$-norms. Moreover, more complex state-of-the-art KD methods [13, 33] using attention-based sampling mechanisms are outperformed, including a +3.5 AP gain using a Faster R-CNN R-50 [21] compared to a vanilla model.

## 1 Introduction

Over the last decade, Convolutional Neural Networks (CNNs), have shown to be a very effective tool in solving fundamental computer vision tasks [14]. One major application of CNNs includes real-time perception systems found in *e.g.* autonomous vehicles, where object detection is often a task of major importance. Deployment of CNNs into real-time applications, however, introduces strict limitations on memory and latency. On the other hand, increased performance of state-of-the-art detectors typically comes with an increase in memory requirements and inference time [12]. Thus, the choice of network model and its according detection performance is strictly limited. Several techniques have been proposed to tackle this problem, *e.g.* pruning [8], weight quantization [9], parameter prediction [6] and Knowledge Distillation (KD) [11]. In this work we are particularly interested in the latter, as it provides an intuitive way of performance improvement without the need for architectural modifications to existing networks.

With KD, the knowledge acquired by a computationally expensive teacher model is transferred to a smaller student model during training. KD has proven to be very effective in tasks such as classification [11], segmentation [19], and in particular has seen considerable progress in detection very recently [5, 7, 13, 33, 35]. Due to the complexity of the output space of a typical detection model, it is necessary to apply KD at the intermediate feature level, as solely relying on output-based KD has proven ineffective [3, 5, 7, 13, 15, 25, 33, 35]. In feature-based KD, in addition to existing objectives, a training objective is introduced which minimizes the *error* between teacher and student activations and is de-facto standard defined by the $\ell_p$-norm distance between individual feature activations [5, 7, 13, 25, 33, 35], as shown in fig. 1a. The $\ell_p$-norm however ignores three important pieces of information present in the feature maps: (i) spatial relationships between features, (ii) the correlation

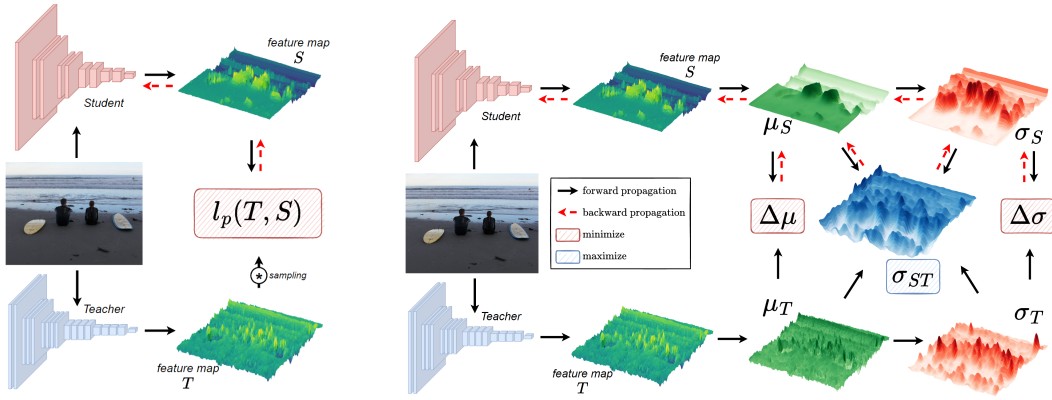

(a) Previous methods [5, 7, 25, 33, 35]. After sampling features an $\ell_p$-norm is applied between selected feature activations.

(b) Our proposed method. We distill relational knowledge in the form of local mean $\mu$, variance $\sigma^2$ and furthermore cross-correlation $\sigma_{\mathcal{ST}}$ between feature spaces.

Figure 1: Feature-based Knowledge Distillation (KD).

between the teacher and student features and (iii) importance of individual features. We notice recent work has (implicitly) focused on bypassing the latter point through mechanisms that sample feature activations by assuming that object regions are more "knowledge-dense" [5, 13, 25, 35]. However, as demonstrated by Guo et al. [7] (2021), even distilling exclusively background feature activations can lead to significant performance improvement, therefore it cannot be assumed that solely object regions contain useful knowledge. Sampling mechanisms furthermore introduce additional drawbacks which may limit their broader implementation into real-world applications, *e.g.* the need for labeled data [7, 13, 25].

In this work, we propose *Structural Knowledge Distillation*, which aims to improve the downsides associated with the $\ell_p$-norm as a central driver for KD methods, rather than designing an ever more sophisticated sampling mechanism. Our key insight is illustrated in fig. 1b: The feature space of a CNN can be locally decomposed into luminance (mean), contrast (variance) and structure (cross-correlation) components, a strategy that has seen successful application in the image domain in the form of SSIM [28]. The new training objective becomes to minimize local differences in mean and variance, and maximize local zero-normalized cross-correlation between the teacher and student activations. Doing so allows us to capture additional knowledge contained in spatial relations and correlations between feature activations of the teacher and additionally the student, rather than directly minimizing the difference in individual activations.

In order to demonstrate the effectiveness of our method we perform extensive experiments using various detection architectures and training schemes. Overall our contribution is as follows:

- We propose *Structural Knowledge Distillation*, which introduces $\ell_{\text{SSIM}}$ and variations as a replacement of the $\ell_p$-norm for feature-based KD in object detection models. This enables the capture of additional knowledge manifested in local mean, variance and cross-correlation relationships in the feature space of student and teacher networks.

- We illustrate through an analysis of the feature space that our method focuses on different areas than $\ell_p$-norms, and that therefore solely sampling from object regions is suboptimal as the entire feature space can contain useful knowledge depending on the activation pattern.

- We demonstrate a consistent quantitative improvement in detection accuracy for various training settings and model architectures by performing extensive experiments on MSCOCO [16]. Our method even performs on par or outperforms carefully tuned state-of-the-art object sampling mechanisms [13, 33], and fundamentally achieves this by only introducing one line of code.

## 2 Related Work

**Knowledge Distillation**   KD aims to transfer knowledge acquired by a cumbersome teacher model to a smaller student model. Bucilǎ et al. [1] (2006) demonstrate that the knowledge acquired by a large ensemble of models can be transferred to a single small model. Hinton et al. [11] (2015) provide a more general solution applied in a DNN in which they raise the temperature of the final softmax until the large model produces a suitably soft set of targets. Most KD research in the computer vision domain focuses on the classification task [24]. However, as our main interest lies in the real-time domain we focus on the more relevant object detection task.

**Object Detection**   Object detection is one of the fundamental computer vision tasks, where speed and accuracy are often two key requirements. Object detectors can be classified into one-stage and two-stage methods, in this work we investigate our approach for both variations. Generally, two-stage detectors allow for higher accuracy at a higher computational cost, and one-stage detectors allow for lower inference times and complexity in exchange for a penalty on accuracy [12]. This notion is however highly dependent on the choice of feature extractor and hyperparameter configuration, which is not a straightforward procedure. The main meta-architecture within the one-stage domain is RetinaNet [17], with extensions including anchor-free modules and Reppoints [37, 32]. In the two-stage domain Faster R-CNN [21] is regarded as the most widely used meta-architecture, where iterations include Cascade R-CNN [2]. Furthermore, regardless of architecture, ResNet [10] backbones are very commonly used to extract features, which are furthermore fused at multiple scales using *e.g.* a FPN [17].

**Knowledge Distillation for Object Detection**   Several methods have been proposed that use KD for object detectors, where it has been found that typically guidance at the intermediate feature level rather than the output is critical due to the complex nature of the output space in detection models [3, 5, 7, 13, 15, 25, 33, 35]. As the detection task requires the identification of multiple objects at different locations, a major complexity introduced is the imbalance between foreground and background, which manifests itself in the intermediate features. Typically, the assumption is made that object regions are "knowledge-dense", and background regions less so. As a result, recent work has implicitly focused on designing mechanisms which sample object-relevant features to distill knowledge from [5, 13, 15, 25, 35].

Li et al. [15] (2017) mimic the features sampled from the region proposals in a two stage detector. Wang et al. [25] (2019) propose imitation masks which locate knowledge dense feature locations based on the annotated boxes. Dai et al. [5] (2021) propose a module which distills based on distance between classification scores. Similarly, Zhixing et al. [35] (2021) use output class probability to determine feature object probability. Recently Kang et al. [13] (2021) proposed a method in which they encode instance annotations in an attention mechanism [23] to locate "knowledge-dense" regions. Contrary to aforementioned methods, Zhang and Ma [33] (2021) propose a purely feature-based method in which they aim to both mimic the attention maps [36] as a sampling mechanism, and furthermore distill through non-local modules [26]. Regardless of the sampling technique, it has been demonstrated by Guo et al. [7] (2021) that it is not necessarily the case that background features are less important for distillation.

**Objective Functions**   The objective in feature-based KD is to minimize the *error* between teacher and student feature spaces during training, typically in addition to existing objectives. The most widely used objective function in feature-based KD is the $\ell_p$-norm with $p = 2$ [3, 5, 7, 15, 25, 33, 35], and less commonly $p = 1$ [3]. The $\ell_p$-norm however, ignores the spatial relationships between features, the correlation between teacher and student and the importance of individual features, of which the latter has been the main focus of previous work. To take into account spatial dependencies, we need to furthermore compare features locally rather than pointwise. SSIM provides an elegant way to take into account spatial dependencies by making local comparisons of intensity and contrast, rather than just pointwise. It is further able to take into account the relationship between the teacher and the student by integrating zero-normalized cross-correlation. Contrary to alternative image signal quality metrics such as VIF [27], GMSD [31] and FSIM [34], SSIM is less complex to formulate mathematically and is differentiable, making it suitable as an objective function.

## 3 Method

### 3.1 Overview

We start off by defining the general form of feature-based distillation loss. For the purposes of this work, we divide a detector into three components: (i) the backbone, used for extracting features, (ii) the neck, for fusing features at different scales (typically a FPN [17]), and (iii) the head, for generating regression and classification scores. For feature-based KD, we select intermediate representations $\mathcal{T} \in \mathbb{R}^{C,H,W}$ and $\mathcal{S} \in \mathbb{R}^{C,H,W}$ from the teacher and student respectively at the output of the neck. The feature-based distillation loss between $\mathcal{T}$ and $\mathcal{S}$ can subsequently be formulated as:

$$\mathcal{L}_{feat} = \sum_{r=1}^{R} \frac{1}{N_r} \sum_{h=1}^{H} \sum_{w=1}^{W} \sum_{c=1}^{C} \mathcal{L}_\varepsilon \left( \nu \left( \phi \left( \mathcal{S}_{r,h,w,c} \right) \right), \nu \left( \mathcal{T}_{r,h,w,c} \right) \right) \tag{1}$$

where $H, W, C, R$ are the height, width, number of channels and number of neck outputs respectively, $N_r = HWC$ the total number of elements for the $r$-th output scale. Additionally we define $\nu(\cdot)$ as a *normalization* function which maps the values of $\mathcal{T}$ and $\mathcal{S}$ to $[0, 1]$, in our case a min-max rescaling layer, and $\phi(\cdot)$ as an optional *adaptation layer* [3] which matches the dimensionality of $\mathcal{T}$ and $\mathcal{S}$, in our case a $1 \times 1$ convolutional layer. We introduce the shortened notation $\mathcal{L}_\varepsilon$ which represents the choice of *difference* measurement function at a single feature position $r, h, w, c$ on normalized features and including the adaptation layer, *i.e.* $\mathcal{L}_\varepsilon = \mathcal{L}_\varepsilon \left( \nu \left( \phi \left( \mathcal{S}_{r,h,w,c} \right) \right), \nu \left( \mathcal{T}_{r,h,w,c} \right) \right)$. Accordingly, we use $\mathcal{S}$ and $\mathcal{T}$ to denote normalized and adapted student and normalized teacher activations respectively, *e.g.* $\mathcal{S} = \nu \left( \phi \left( \mathcal{S}_{r,h,w,c} \right) \right)$.

### 3.2 Measuring Difference

As ascertained, the de-facto standard choice for $\mathcal{L}_\varepsilon$ is the $\ell_p$ norm. $p = 2$ penalizes large errors, but is more tolerable to smaller errors. On the other hand, $p = 1$ does not over-penalize large errors, but smaller errors are penalized more harshly. The $\ell_p$ norm in its general form is given by:

$$\ell_p : \mathcal{L}_\varepsilon = \left( |\mathcal{S} - \mathcal{T}|^p \right)^{1/p} \tag{2}$$

Clearly such a function is not able to capture spatial relationships between features. In order to capture second-order information we need to involve at least two feature positions, we therefore change the problem statement from a *point-wise* comparison to a local *patch-wise* comparison. For each such patch, we extract three fundamental properties: the mean $\mu$, the variance $\sigma^2$, and the cross-correlation $\sigma_{\mathcal{S}\mathcal{T}}$ which captures the relationship between $\mathcal{S}$ and $\mathcal{T}$. We follow [28] and compute these quantities using a Gaussian-weighted patch $F_{\sigma_F}$ of size $11 \times 11$ and $\sigma_F = 1.5$. The proposed SSIM framework [28] compares each of the properties, and is therefore composed of three components: luminance $l$, contrast $c$ and structure $s$, which are defined as follows:

$$l = \frac{2\mu_{\mathcal{S}}\mu_{\mathcal{T}} + C_1}{\mu_{\mathcal{S}}^2 + \mu_{\mathcal{T}}^2 + C_1} \quad \text{(3a)} \qquad c = \frac{2\sigma_{\mathcal{S}}\sigma_{\mathcal{T}} + C_2}{\sigma_{\mathcal{S}}^2 + \sigma_{\mathcal{T}}^2 + C_2} \quad \text{(3b)} \qquad s = \frac{\sigma_{\mathcal{S}\mathcal{T}} + C_3}{\sigma_{\mathcal{S}}\sigma_{\mathcal{T}} + C_3} \quad \text{(3c)}$$

where $\mu_{\mathcal{S}}, \mu_{\mathcal{T}}$ refer to the mean, $\sigma_{\mathcal{S}}, \sigma_{\mathcal{T}}$ refer to the variance and $\sigma_{\mathcal{S}\mathcal{T}}$ refers to the covariance within the patch. Furthermore, to prevent instability $C_1 = (K_1 L)^2$, $C_2 = (K_2 L)^2$, $C_3 = C_2/2$, where $L$ is the dynamic range of the feature map and $K_1 = 0.01$, $K_2 = 0.03$. An important property of eq. (3) is that it assigns more importance to *relative* changes in $l$ and $c$ due to the quadratic terms in the denominator. Furthermore, $s$ is a direct measurement of the zero-normalized correlation coefficient between $\mathcal{S}$ and $\mathcal{T}$, and hence is formulated as the ratio between their covariance and product of standard deviations. As the range of eq. (3) is $[-1, 1]$, combining the three components $l, c, s$ results in the following objective:

$$\ell_{\text{SSIM}} : \mathcal{L}_\varepsilon = (1 - \text{SSIM})/2 = \left( 1 - \left( l^\alpha \cdot c^\beta \cdot s^\gamma \right) \right)/2 \tag{4}$$

where the prevalence of each function can be tuned, with $\alpha = \beta = \gamma = 1.0$ as a default. As our method is purely feature-based and therefore independent of the type of head or bounding box labels, we simply add $\mathcal{L}_{feat}$ to the existing detection objective function $\mathcal{L}_{det}$ (typically $\mathcal{L}_{cls}$ and $\mathcal{L}_{reg}$) using weighting factor $\lambda$, which results in the following overall training objective:

$$\mathcal{L} = \lambda \mathcal{L}_{feat} + \mathcal{L}_{det} \tag{5}$$

# 4 Experiments

## 4.1 Experiment Settings

Following literature [7, 13, 25, 33, 35], we assess the performance of $\ell_{\text{SSIM}}$ on the MSCOCO [16] validation dataset. We report mean Average Precision (AP) as the main evaluation metric, and additionally report AP at specified IoU thresholds $AP_{50}, AP_{75}$ and object sizes $AP_S, AP_M, AP_L$. Our central points of comparison are the two most widely used one- and two-stage meta-architectures, RetinaNet (RN) [18] and Faster-RCNN (FRCNN) [21]. We use ResNet/ResNeXt-101 backbones [10, 30] for the teachers and R-50 [10] backbones for all students, with $\lambda = 4, 2$ respectively. We conduct our experiments in Pytorch [20] using the MMDetection2 [4] framework on a Nvidia RTX8000 GPU with 48GB of memory. Each model is trained using SGD optimization with momentum 0.9, weight decay 1e-4 and batch size 8. The learning rate is set at 0.01 (RN) / 0.02 (FRCNN) and decreased tenfold at step 8 and 11, for a total of 12 epochs. We additionally implement batch normalization layers after each convolutional layer, and use focal loss [18] with $\gamma_{fl} = 2.0$ and $\alpha_{fl} = 0.25$. The input images are resized to minimum spatial dimensions of $800$ while retaining the original ratios, and we add padding to both fulfill the stride requirements and retain equal dimensionality across each batch. Finally the images are randomly flipped with $p = 0.5$ and normalized.

## 4.2 Comparison with $\ell_p$-norms

In this first set of experiments we compare the performance of $\ell_p$-norms to $\ell_{\text{SSIM}}$. Table 1 shows the results of the main experiments comparing the best performance of each $\mathcal{L}_\varepsilon$. It can be observed that: (i) $\ell_{\text{SSIM}}$ outperforms $\ell_p$-norms by a significant margin, boosting performance with up to +3.7AP. (ii) Adopting any form of feature-based distillation results in an improvement over the vanilla network, except in Faster R-CNN [21]. (iii) Even though previous work uses $\ell_2$, $\ell_1$ outperforms $\ell_2$ with AP improvements of 2.3 vs. 0.4 and 1.2 vs. 0.0 respectively.

Table 1: Comparison of objective functions on MSCOCO [16].

| Backbone | $\mathcal{L}_\varepsilon$ | AP | $AP_{50}$ | $AP_{75}$ | $AP_S$ | $AP_M$ | $AP_L$ |
|---|---|---|---|---|---|---|---|
| | | RetinaNet [18] | | | | | |
| Teacher R101 | | 41.0 | 60.3 | 44.0 | 24.1 | 45.3 | 53.8 |
| *Vanilla R50* | | *36.4* | *55.6* | *38.7* | *21.1* | *40.3* | *46.6* |
| R50 | $\ell_2$ | 36.8 (+0.4) | 55.7 | 39.1 | 20.6 | 40.5 | 47.3 |
| R50 | $\ell_1$ | 38.7 (+2.3) | 57.6 | 41.6 | 22.7 | 42.7 | 50.5 |
| **R50** | $\ell_{\text{SSIM}}$ | **40.1 (+3.7)** | **59.2** | **43.1** | **23.1** | **44.6** | **53.2** |
| | | Faster R-CNN R50 [21] | | | | | |
| Teacher X-101 | | 45.6 | 64.1 | 49.7 | 26.2 | 49.6 | 60.0 |
| *Vanilla R50* | | *37.4* | *58.1* | *40.4* | *21.2* | *41.0* | *48.1* |
| R50 | $\ell_2$ | 37.4 (+0.0) | 57.6 | 40.9 | 21.2 | 41.3 | 48.1 |
| R50 | $\ell_1$ | 38.6 (+1.2) | 58.8 | 42.1 | 21.8 | 42.1 | 49.9 |
| **R-50** | $\ell_{\text{SSIM}}$ | **40.9 (+3.5)** | **61.0** | **44.9** | **23.7** | **44.5** | **53.5** |

To investigate what this improvement in performance can be attributed to, we analyze the distribution of the training stimulus in the feature space. Figure 2 illustrates the comparison between $\ell_{\text{ssim}}$ and $\ell_p$ ($p = 2$) for the magnitude of the loss, averaged over all channels and 12 training epochs in neck $r = 1$. This tells us something about which regions in the feature space are focused on more. It can be observed that with $\ell_2$, high loss is assigned to object regions in particular, and furthermore regions with high brightness, such as the sky in fig. 2a or the window in fig. 2b. $\ell_{\text{ssim}}$ however assigns the loss differently, where not only object regions are focused, but additionally more diverse background regions are targeted, while little importance is given to low-contrast background regions.

One of the issues highlighted by [7] is that losses are higher in object regions than in background regions. As can be seen in fig. 2, the loss applied by $\ell_{\text{ssim}}$ is much more distributed over the feature space than $\ell_2$, which as a direct result causes a more distributed application of the gradient in the feature space. As a result, is can be observed that the feature map of a $\ell_{\text{ssim}}$ distilled model is much

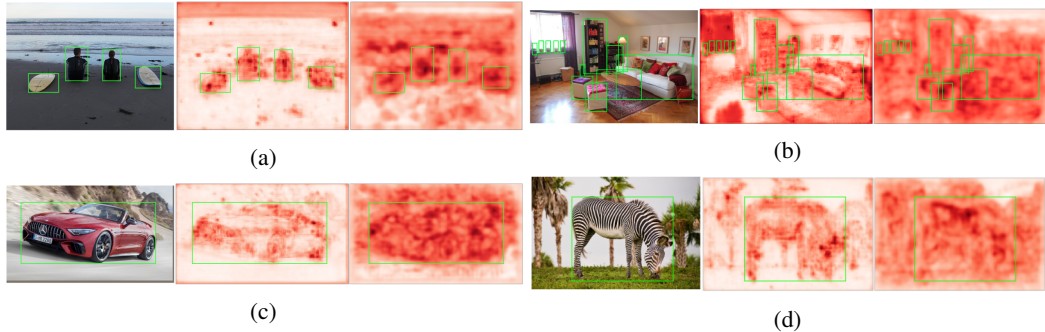

(a)                  (b)

(c)                  (d)

Figure 2: Distribution of the magnitude of the loss in the feature space at output scale $r = 1$ for various images, averaged over all channels and 12 epochs of training on MSCOCO [16]. From left to right: Image, student trained with $\ell_2$, student trained with $\ell_{\text{ssim}}$. A darker color indicates a higher loss, object regions have been highlighted with bounding boxes, and feature maps have been normalized.

more similar to the teacher than an $\ell_p$ distilled model, as shown in fig. 3, which directly translates to the increase in performance.

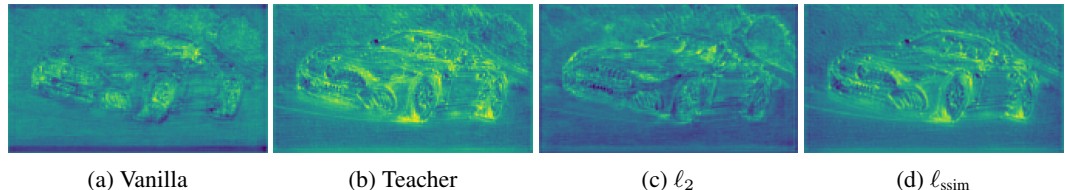

(a) Vanilla      (b) Teacher      (c) $\ell_2$      (d) $\ell_{\text{ssim}}$

Figure 3: Qualitative comparison of a channel sampled randomly from RetinaNet [18] intermediate neck output scale $r = 1$. Lighter colors indicate higher activation values.

## 4.3 Influence of Luminance, Contrast and Structure

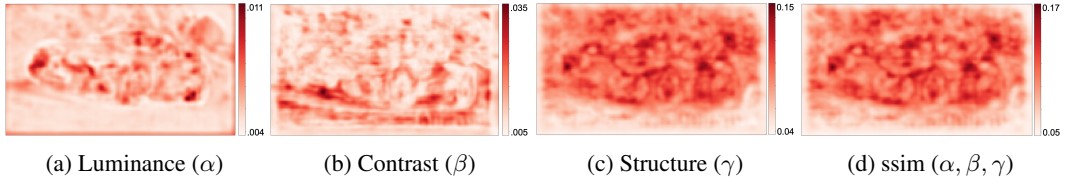

(a) Luminance ($\alpha$)   (b) Contrast ($\beta$)   (c) Structure ($\gamma$)   (d) ssim ($\alpha, \beta, \gamma$)

Figure 4: Distribution of the channel averaged magnitude of the loss for each individual component luminance, contrast and stucture, averaged over 12 epochs of training on MSCOCO [16].

Next we compare the influence of the luminance, contrast and structure components by tuning $\alpha$, $\beta$ and $\gamma$ respectively (refer to eq.(4)), for this experiment we use the RetinaNet [18] teacher-student pair. The results are shown in table 2. The main observation that can be made here is that the influence of structure ($\gamma$) is more substantial than the other components, and even on its own provides an increase of +3.2 AP. Furthermore, the comparisons of luminance $\alpha$ and contrast $\beta$ alone result in performance comparable or better than $\ell_p$-norms (compare to table 1). In fig. 4 we furthermore compare the average magnitude of the loss during training. The luminance provides a similar training stimulus to the $\ell_p$-norm (ref. to fig. 2c), but is more "smooth" due to the Gaussian kernel. Contrarily, contrast mostly targets background areas, as it is more sensitive to differences where base contrast is already low. Finally, it can be noticed that structure has the most influence over the total loss, both in magnitude and spatial distribution.

Table 2: Comparison of objective functions for RetinaNet [18] on MSCOCO [16]. $\alpha$ tunes luminance, $\beta$ tunes contrast and $\gamma$ tunes structure.

| Backbone | $\alpha$ | $\beta$ | $\gamma$ | AP | $AP_{50}$ | $AP_{75}$ | $AP_S$ | $AP_M$ | $AP_L$ |
|---|---|---|---|---|---|---|---|---|---|
| Teacher ResNet-101 | | | | 41.0 | 60.3 | 44.0 | 24.1 | 45.3 | 53.8 |
| *Vanilla ResNet-50* | | | | *36.4* | *55.6* | *38.7* | *21.1* | *40.3* | *46.6* |
| ResNet-50 | 1 | 0 | 0 | 38.7 (+2.3) | 57.9 | 41.6 | 21.8 | 42.8 | 50.8 |
| ResNet-50 | 0 | 1 | 0 | 38.9 (+2.5) | 57.7 | 41.6 | 21.7 | 42.6 | 51.3 |
| ResNet-50 | 0 | 0 | 1 | 39.6 (+3.2) | 58.6 | 42.7 | 22.5 | 44.0 | 52.5 |
| ResNet-50 | 0 | 1 | 1 | 40.0 (+3.6) | 59.0 | 42.8 | 22.4 | 44.4 | **53.3** |
| **ResNet-50** | 1 | 1 | 1 | **40.1 (+3.7)** | **59.2** | **43.1** | **23.1** | **44.6** | 53.2 |

Additionally, in fig. 5 we illustrate the differences between student and teacher activations for the differently trained models. It can be observed that the structure objective results in a feature space that has converged to a very similar local optimum as the teacher, with few noisy or large differences. Luminance contains more noisy differences, and especially in the last layer demonstrates high differences. Although the contrast objective performs relatively well on its own, it seems to provide different activations as the teacher, particularly in the object area, as the student network converged to a different local minimum.

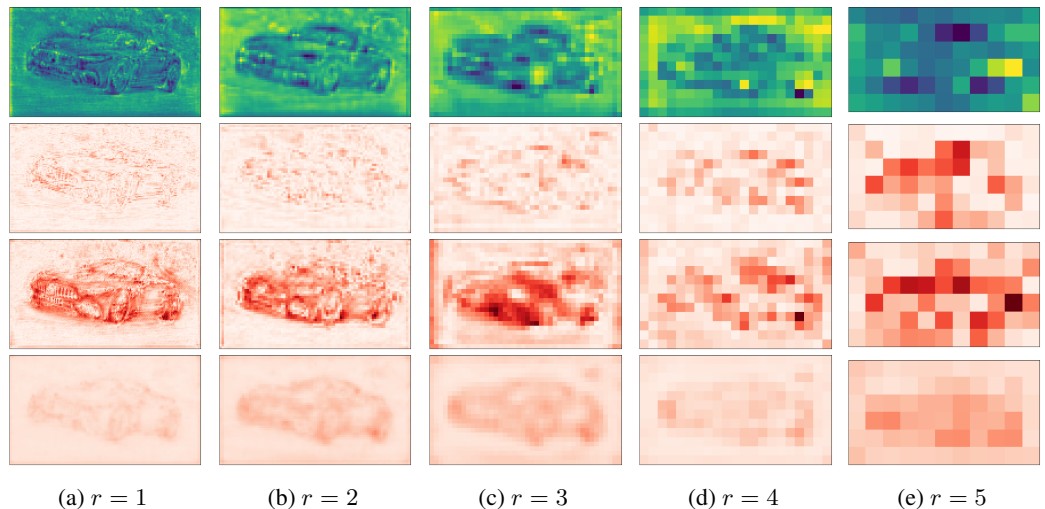

    (a) $r = 1$      (b) $r = 2$      (c) $r = 3$      (d) $r = 4$      (e) $r = 5$

Figure 5: Top row: Channel averaged activations in the RetinaNet R101 [18] Teacher. Subsequent rows illustrate channel averaged differences in activations between teacher and student, distilled with: 2nd row: luminance ($\alpha$). 3rd row: contrast ($\beta$). 4th row: structure ($\gamma$). $r$ represents the output scales of the feature map (eq. 1). Differences have been normalized, where darker color indicates a higher value.

## 4.4 Comparison to State-of-the-Art Methods

Next, we compare to recent work, for which the following methods serve as baselines: (i) Zhang and Ma [33] (2021), a purely feature-based approach leveraging attention masks [36] and non-local modules [26], and (ii) Kang et al. [13] (2021), who encode labeled instance annotations in an attention mechanism [23] and report state-of-the-art for distillation methods for RetinaNet [18] and Faster R-CNN [21] on MSCOCO [16] at the time of writing. To further demonstrate the simplicity and versatility of our method, we use the original code and teachers as the authors to compare to our proposed method in the same experimental setup. [33] use the same MMDetection2 [4] framework, while [13] use Detectron2 [29]. For the comparison with [13] we furthermore adopt inheritance, a practice proposed by the authors in which the FPN [17] and head of the student are initialized with teacher parameters. This leads to faster training convergence, but may not be applicable when

architectures differ between teacher and student. As the teachers and exact configurations slightly vary, we split up the comparison into two parts, as shown in table 3.

Table 3: Comparison to state-of-the-art methods on MSCOCO [16]. † denotes inheritance.

| Method | RetinaNet [18] | | | | Faster R-CNN [21] | | | |
|---|---|---|---|---|---|---|---|---|
| | AP | $AP_S$ | $AP_M$ | $AP_L$ | AP | $AP_S$ | $AP_M$ | $AP_L$ |
| Teacher | 41.0 | 24.1 | 45.3 | 53.8 | 45.6 | 26.2 | 49.6 | 60.0 |
| *Vanilla* | *36.4* | *21.1* | *40.3* | *46.6* | *37.4* | *21.2* | *41.0* | *48.1* |
| Zhang and Ma [33] | 38.5 (+2.1) | 21.7 | 42.6 | 51.5 | 38.9 (+1.5) | 21.9 | 42.1 | 51.5 |
| **Ours** | **40.1 (+3.7)** | **23.1** | **44.6** | **53.2** | **40.9 (+3.5)** | **23.7** | **44.5** | **53.5** |
| Teacher | 40.4 | 24.0 | 44.3 | 52.2 | 42.0 | 25.2 | 45.6 | 54.6 |
| *Vanilla* | *37.4* | *23.1* | *41.6* | *48.3* | *37.9* | *22.4* | *41.1* | *49.1* |
| **Kang et al. [13]** † | **40.7 (+3.3)** | **24.6** | 44.9 | 52.4 | 40.9 (+3.0) | **24.5** | 44.2 | 53.3 |
| **Ours** † | **40.7 (+3.3)** | 24.0 | **45.0** | **53.1** | **41.0 (+3.1)** | 23.8 | **44.5** | **53.7** |

It can be observed that: (i) the adoption of our $\ell_{\text{SSIM}}$ as the distillation function results in an improvement of +3.7 AP, and outperforms [33] for all box sizes and IoU thresholds. (ii) Both our $\ell_{\text{SSIM}}$ and [13] result in an improvement of +3.3 AP over the vanilla network. In particular, our method scores high for $AP_L$, while [13] mainly show better performance in the small object $AP_S$ category. Additionally it can be observed that the student is able to outperform the teacher with RetinaNet [18].

## 4.5 Ablation Studies

**Generalizability to Detection Architectures and Schedules** We perform additional studies on several different detection architectures to demonstrate the generalizability of our method. We evaluate our distillation method on the smaller ResNet-18 backbone and two alternative one-stage architectures, Fsaf-RetinaNet [37], which extends RetinaNet [18] with an anchor-free module, and Reppoints [32], which replaces the regular bounding box representation of objects by a set of sample points. The results of our experiments are shown in table 4. It can be observed that: (i) For each detection architecture our method significantly improves performance, with +3.5AP for the ResNet18 backbone, +2.3 AP for Fsaf-RetinaNet [37], +3.3 AP for Reppoints [32]. (ii) In general, our method is modular and can significantly improve performance regardless of the detection framework used.

Table 4: Investigation of several popular detection architectures on MSCOCO [16].

| Model | AP | $AP_{50}$ | $AP_{75}$ | $AP_S$ | $AP_M$ | $AP_L$ |
|---|---|---|---|---|---|---|
| RetinaNet-R101 (Teacher) | 41.0 | 60.3 | 44.0 | 24.1 | 45.3 | 53.8 |
| RetinaNet-R18 (Vanilla) | 32.6 | 50.6 | 34.6 | 17.8 | 35.2 | 43.5 |
| **RetinaNet-R18 (Ours)** | **36.1 (+3.5)** | **54.3** | **38.6** | **18.9** | **39.7** | **49.2** |
| RetinaNet-R101 (Teacher) | 41.0 | 60.3 | 44.0 | 24.1 | 45.3 | 53.8 |
| RetinaNet-R50 (Vanilla, 2x) | 37.4 | 56.7 | 39.6 | 20.0 | 40.7 | 49.7 |
| **RetinaNet-R50 (Ours, 2x)** | **40.6 (+3.2)** | **59.7** | **43.7** | **23.6** | **44.8** | **53.9** |
| Fsaf-RetinaNeXt-X101 [37] (Teacher) | 42.4 | 62.5 | 45.5 | 24.6 | 46.1 | 55.5 |
| Fsaf-RetinaNet-R50 [37] (Vanilla) | 37.4 | 56.8 | 39.8 | 20.4 | 41.1 | 48.8 |
| **Fsaf-RetinaNet-R50 [37] (Ours)** | **39.7 (+2.3)** | **59.3** | **42.4** | **22.0** | **43.3** | **52.0** |
| Reppoints X-101[32] (Teacher) | 44.2 | 65.5 | 47.8 | 26.2 | 48.4 | 58.5 |
| Reppoints-R50 [32] (Vanilla) | 37.0 | 56.7 | 39.7 | 20.4 | 41.0 | 49.0 |
| **Reppoints-R50 [32] (Ours)** | **40.3 (+3.3)** | **60.3** | **43.5** | **22.6** | **44.4** | **53.9** |

**Effects of an Adaptation Layer** Adaptation layers can be implemented when channel or spatial dimensions between teacher and student do not match, and have shown to generally improve performance in previous methods [3, 25, 33]. We implement the commonly used $1 \times 1$ convolution to investigate the influence on performance with our method. We adopt the RetinaNet R101-50

teacher-student pair and the CRCNN X101 - FRCNN R50 teacher-student pair. The results are shown in table 5. We notice that in table 5a there is no additional benefit of adopting an adaptation layer, while in table 5b the difference is significant, and implementing the adaptation layer is critical. Our method can therefore be used both with and without adaptation. However, when architectures and backbones differ between teacher and student the adaptation layer is highly beneficial.

Table 5: Investigation of the effect of adaptation layers

(a) RetinaNet R101 - R50 [18]

| Adap. layer | AP | $AP_S$ | $AP_M$ | $AP_L$ |
|---|---|---|---|---|
| none | **40.1** | **23.1** | **44.6** | 53.2 |
| $1 \times 1$ | **40.1** | **23.1** | 44.4 | **53.4** |

(b) Cascade RCNN X101 [2] - FRCNN R50 [21]

| Adap. layer | AP | $AP_S$ | $AP_M$ | $AP_L$ |
|---|---|---|---|---|
| none | 39.8 | 22.6 | 43.4 | 52.1 |
| $1 \times 1$ | **40.9** | **23.7** | **44.5** | **53.5** |

**Varying Patch Size $F$ and Loss Prevalence $\lambda$**  We additionally investigate the two remaining main hyperparameters that we introduce in this work, for which we use the RetinaNet R101-50 [18] teacher-student pair. The influence of the prevalence of $\mathcal{L}_{feat}$ tuned by $\lambda$ is shown in fig. 6, where it can be noticed that the choice of $\lambda$ can cause a difference of up to +0.5 AP, with $\lambda = 2, 4$ providing the best performance. Additionally the influence of the local patch size $F$ over which we calculate each component of $\ell_{\text{ssim}}$ is investigated, the results are shown in fig. 6. It can be noticed that the choice of kernel size does not have significant influence over performance.

**Error Types**  Finally we are interested in the type of improvements made by our distillation method, as illustrated in fig. 7. We can particularly observe improvements in the $AP_L$ category (rightmost set of columns), which can also be noticed in table 3 where we perform better in this category than previous methods. One explanation is illustrated in fig. 5d, 5e where we can see that the structural part ensures convergence towards the teacher, particularly in the deeper layers which are responsible for detecting large objects. We furthermore observe that the main performance improvements take place in localization, as seen in the $AP_{75}$, $AP_{50}$ and $AP_{10}$ (Loc) categories in fig. 7. The remaining categories refer to performance evaluation after removal of class supercategories (Sim), all classifications (Oth) and all background FP's, where across the board improvement remains more limited than in the localization categories.

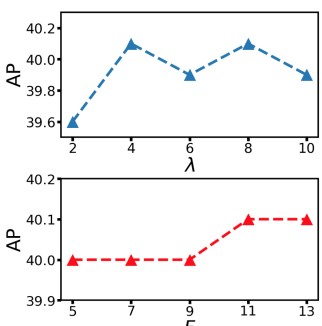

Figure 6: Performance difference when varying hyperparameters. Top: KD loss prevalence $\lambda$. Bottom: kernel size $F$.

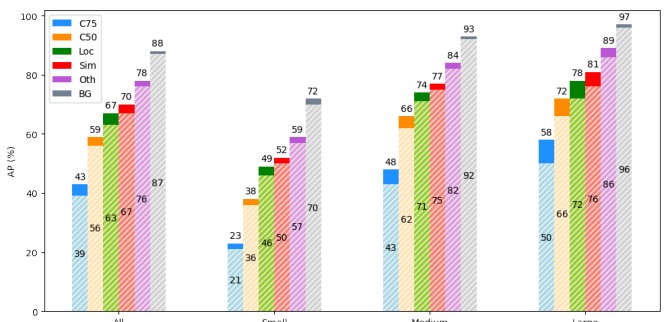

Figure 7: RetinaNet R-50 [18] AP score for varying box sizes. Hatched areas represent the vanilla model, solid areas represent the performance increase obtained through our distillation method.

# 5 Conclusion

This paper proposed $\ell_{\text{ssim}}$, a replacement for the conventional $\ell_p$-norm as a building block for feature-based KD in object detection. By taking into account additional contrast and structural cues, feature importance, correlation and spatial dependence are considered in the loss formulation. $\ell_{\text{ssim}}$ outperforms $\ell_p$-norms by a great margin and is able to reach performance on par or even surpass state-of-the-art without the need for carefully designed and complex sampling mechanisms. Our method is simple and can be implemented by replacing one line of code. We propose three main directions for future work: First, using $\ell_{\text{ssim}}$ as a building block for future KD methods within the object-centric vision domain. Second, integrating and modifying $\ell_{\text{ssim}}$ to work in other types of (vision) tasks. Finally, an investigation from a theoretical point-of-view on the impact of convergence trajectories and optimization performance of loss functions, not limited to those presented in this paper, as applied in the feature space of a DNN.

## Reproducibility Statement

All our experiments are based on publicly available frameworks [4, 29] and datasets [16]. An example implementation of KD loss between teacher and student features is shown below. Omitting the import and using a library such as Kornia [22], a change from $\ell_2$ to $\ell_{\text{SSIM}}$ only requires a change in **one line of code**.

```
─────────────────────────── l2 implementation ───────────────────────────
from torch.nn.functional import mse_loss
def kd_loss(student_feats, teacher_feats):
    # inputs have shape [B, C, H, W]
    kd_feat_loss = mse_loss(student_feats, teacher_feats)
    return kd_feat_loss
```

```
─────────────────────────── ssim implementation ───────────────────────────
from kornia.losses import ssim_loss
def kd_loss(student_feats, teacher_feats):
    # inputs have shape [B, C, H, W]
    kd_feat_loss = ssim_loss(student_feats, teacher_feats, window_size=11)
    return kd_feat_loss
```

## Acknowledgements

The research leading to these results is funded by the German Federal Ministry for Economic Affairs and Climate Action within the project "KI Delta Learning" (Förderkennzeichen 19A19013A). The authors would like to thank the consortium for the successful cooperation.

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
