# OpenReview forum: "Structural Knowledge Distillation for Object Detection"
_NeurIPS.cc/2022/Conference — NeurIPS 2022 Accept_

### Official Review · Reviewer_XGaw · 2022-06-29

**Rating:** 8
**Confidence:** 3
**Soundness:** 4 excellent
**Presentation:** 4 excellent
**Contribution:** 4 excellent

**Summary:**

This paper focuses on knowledge distillation for object detection. Authors conclude the limitations of conventional feature based knowledge distrillation from three aspects, and propose novel distillation strategy to solve these challenges. Particularly, it performs knowledge distillation on patch-level features instead of existing point-level features. It estimates mean, variance, and covariance on student and teacher feature maps, and compute three terms to formulate the knowledge distillation learning objective. The experimental results support the analysis in the paper and demonstrate the effectiveness of proposed method.

**Questions:**

My questions have been written in the weakness part.

**Limitations:**

No obvious negative social impact.

**Strengths And Weaknesses:**

Strengths:
1. The analysis of conventional feature based knowledge distillation makes sense, and the modified patch-level knowledge distillation is well motivated and can reflect the internal spatial and structural information.
2. The proposed new formulation is simple and easy to follow, it has potential to boost general knowledge distillation in computer vision tasks with little extra computation consumption.
3. The performance improvement is significant. The ablation studies well support that the structural information is crucial in konwledge distillation, and it can enlight new research directions in this area.

Weaknesses:
1. How will the performance change if replace original knowledge distillation objective in [14] with $l_{ssim}$?

Overall, from my perspective, this paper is well written and the method is simple and generalizable. I did not see significant weakness in this work.

---

> ### Author Response · Authors · 2022-08-02
> **Response to reviewer XGaw**
>
> We thank the reviewer for their time and are pleased to hear that our work has been received favourably.
> The response to the question posed by the reviewer can be found in the general comment addressing similar questions from multiple reviewers.

---

### Official Review · Reviewer_cG4E · 2022-07-10

**Rating:** 6
**Confidence:** 5
**Soundness:** 2 fair
**Presentation:** 3 good
**Contribution:** 2 fair

**Summary:**

Instead of minimizing the feature distance directly, the paper distills student models by minimizing three components: luminance, contrast, and feature structure, which are expressed by the mean, variance, and covariance between feature patches. It is verified on MS-COCO with two detection frameworks, e.g., RetinaNet and Faster R-CNN.


**Questions:**

- In the feature space, what's the meaning of luminance?
- If the distillation method doesn't depend on the heads, is it possible to extend it to a more general method that can be used for classification, instance segmentation (Mask R-CNN)?
- Please update Figure 1 using a high-resolution version.

**Limitations:**

None.

**Strengths And Weaknesses:**

Strengths:

- Writing is pretty well-formulated and well-organized. It’s easy to follow.
- Interesting idea that proposes three components in feature space.
- The proposed method doesn't depend on the detection head or other parts.

---

> ### Author Response · Authors · 2022-08-02
> **Response to reviewer cG4E**
>
> We thank the reviewer for their time, and would like to address the questions and comments below.
>
> ## Questions
>
> ### Q. In the feature space, what's the meaning of luminance?
> A. We followed the standard terms of ssim in this context and  defined it in eq. (3a). It represents the relative comparison of the mean activation intensity of the local patches of teacher and student feature maps T and S. It is defined per feature channel, thus, luminance reflects the activation intensity of a single feature at an image position.
>
> ### Q. If the distillation method doesn't depend on the heads, is it possible to extend it to a more general method that can be used for classification, instance segmentation (Mask R-CNN)?
> A. This question has been adressed in the general comment to all reviewers, as it showed a degree of similarity with a question posed by another reviewer.
>
> ### Q. Please update Figure 1 using a high-resolution version.
> A. Thank you for pointing this out, we will replace the figure with a higher resolution one in the camera ready version.

---

### Official Review · Reviewer_VXDw · 2022-07-11

**Rating:** 7
**Confidence:** 5
**Soundness:** 3 good
**Presentation:** 2 fair
**Contribution:** 3 good

**Summary:**

1. In this paper the author proposed SSIM loss for object detection distillation and verified it's improvement for various training settings and model architectures. In this paper, feature map was devided into small patched with gaussian smooth, loss was computed per patch. In each patch, mean, variance and convariance were computed and SSIM loss composed of there weighted product.
2. This paper showed SSIM loss can give more siganl(especially from background part) in model training and lead to a better optimization. Also, SSIM loss focuses on differentareas than lp -norms

**Questions:**

1. Give more detailed analysis to help readers understand how SSIM loss help  train better object detection model.
2. Mention all concepts in related work part.

**Limitations:**

Yes the author adequately addressed the limitations and potential negative societal impact.

**Strengths And Weaknesses:**

Strengths
1. Propose a simple and effective way to boost the performance of  knowledge distillation, especially on object detection.
2. Did enough experiments to illustrate that SSIM loss works for different object detection models and teacher-student pair.
3. Provide Insight why SSIM loss works better than l1/l2 loss and showed which component is the most effective.
 Weaknesses
1. The paper is not easy to follow for ones who is not familiar with object detection or distillation. For example, the author mentioned "r = 1" in line 180 and fig 2,3,5. However author  does not tell it refers to different layer in FPN.
2. There are some interesting phenomenon, such as SSIM provide higher loss in learning, distillation contribute more on bounding box location accuracy and adaption layer works in some setting but does not work in others. However, the authors failed to provide insights, certainly doesn't provide theory to explain this. Thus, this paper looks more like a tech report rather than a conference paper.
3. The SSIM loss works directly on feature maps, does not change object detection heads or losses. The methods proposed in the paper should works in all vision tasks such as classification, segmentation or recognition. However this is not mentioned in the paper.

---

> ### Author Response · Authors · 2022-08-02
> **Response to reviewer VXDw**
>
> We thank the reviewer for their time and thorough review, and are pleased to hear that our work has been received favourably.
> In particular, we are grateful for mentioning the missing explanation of the parameter "r". In eq. (1) and line 124, R (and indirectly r) are defined as the neck output levels of the detector, it is indeed a good point to explicitly state that it concerns the output of the FPN, which we use as the "neck" in our experiments. We will adjust this for the camera-ready version.
>
> The comment regarding the application of our method to additional tasks is addressed in the general comment to all authors, as it had similarity with a question from another reviewer. We address the remaining questions below.
>
> ### Questions
> #### Q. Give more detailed analysis to help readers understand how SSIM loss help train better object detection model.
> We answer this question by explaining our reasoning, which is twofold. First, we tried to provide a good intuition and motivation to use SSIM in the introduction. Second, we added both qualitative examples and quantitative ablation studies in order to provide these insights. Through these insights, we aim to demonstrate our method's ability to take into account knowledge contained in spatial relationships between features, which is something that L_p-norms lack. These results raise new questions to how we apply distillation in most sota works, which is why we want to make these insights available to the community to further build on.
>
> #### Q. Mention all concepts in related work part.
> We focused on knowledge distillation on object detection in related work due to space limitations. We will mention knowledge distillation concepts for different tasks such as segmentation or recognition in the first paragraph of related work for completeness.

---

### Official Review · Reviewer_6uNd · 2022-07-11

**Rating:** 5
**Confidence:** 5
**Soundness:** 3 good
**Presentation:** 3 good
**Contribution:** 2 fair

**Summary:**

This paper proposes Structural Knowledge Distillation (SKD) as a replacement for the pixel-wise independent Lp norm. The proposed method is motivated with SSIM to take into account additional contrast and structural cues to preserve more information. Extensive experiments on MSCOCO demonstrate the effectiveness of this method across different training schemes and architectures.

**Questions:**

question:
1.In line 124, can you explame what normalization function you use to limit the value of feature maps to [0, 1]?
2.In the field of image classification, most KD methods based on feature can do a better performance with the combination of logits-based KD methods. I notice that your methods totally based on feature map, did or will you do some experiments on combining your distillation method with other distillation method?

**Limitations:**

1. The explanation and theory analysis of the proposed method is limited.
2. The compared SOTA methods is not enough.

**Strengths And Weaknesses:**

strength:
1.First use ssim as the function to decribe the difference between teacher and student feature maps.
2.Simple method design with good result improvement over the vanilla model.
3.Nice ablation experiment to show the effectiveness of Structural Knowledge Distillation.

weakness:
1.There is not enough theory in this article to explain the effectiveness of Structural Knowledge Distillation.
2.In sec4.4, only two SOTA KD methods in detections used for comparing.

---

> ### Author Response · Authors · 2022-08-02
> **Response to Reviewer 6uNd - Limitations**
>
> We thank the reviewer for their time and for providing a thorough review. We start off with addressing some of the general limitations stated by the reviewer, after which we will address specific questions in a separate comment.
>
> ### Comparison to more SOTA methods
> One main concern about the paper is the amount of comparisons to SOTA methods in table 3 in the paper, where we directly re-evaluate and report results for two methods [14, 33] with two detection architectures (RetinaNet and Raster R-CNN).
> The choice for these particular two methods was very deliberate as they are representative for the two most important lines of work.
>
> [33] reflects the current state-of-the-art performance within **purely feature-based** methods (c.f. lines 210-212), to which also our approach belongs. One additional purely feature-based method is Heo et al. (ICCV 2019), which we chose not to re-evaluate as the work of [33] already did so and yielding limited performance. This is shown in table 1 below, where the reported numbers of each method are listed.
>
>
> | Method                   | TS | Teacher AP | Vanilla AP | Student AP  |
> |--------------------------|----|------------|------------|-------------|
> | Heo at al. (ICCV 2019)   | 2x | 41.0       | 37.4       | 37.8 (+0.4) |
> | [33] Zhang et al. | 2x | 41.0  | 37.4       | 39.6 (+2.2) |
> | [33] Zhang et al. | 1x | 41.0  | 36.4       | 38.5 (+2.1) |
> | Ours | 2x | 41.0       | 37.4       | 40.6 (+3.2) |
> | Ours | 1x | 41.0       | 36.4       | 40.1 (+3.7) |
> #### Table 1: Average precision (AP) of **purely feature-based** object detection methods for RetinaNet as reported in [33]. TS: Training Schedule, 1x = 12 epochs, 2x = 24 epochs.
>
>
>
> [14] is the state-of-the-art method for object detection knowledge distillation **without restriction** on purely feature-based methods, c.f. lines 212-214 and the table below. [8, 14, 34] have similar training settings (2x schedule), with a slightly different teacher, while [14] yields best results. We discarded [5], as they implemented their method in a proprietary framework, and their reported results are worse than ours. Additionally note that we also did not re-evaluate [3, 25] (c.f. line 86), as their reported performance was already significantly surpassed by the methods mentioned in table 1 and 2, (e.g. [8, 14, 33, 34]) and their methods are only applicable to a limited range of architectures. Overall, we found [14] to be best suited as baseline.
>
>
>
> | Method                        | FB  | framework    | C   | TS  | Teacher AP | Vanilla AP | Student AP  |
> |-------------------------------|-----|--------------|-----|-----|------------|------------|-------------|
> | [34] Zhixing et al. | no  | mmdetection2 | yes | 2x  | 38.9       | 37.4       | 39.7 (+2.3) |
> | [8] Guo et al.         | no  | mmdetection2 | no  | 2x* | 40.5       | 36.5       | 39.7 (+3.2) |
> | [34] Zhixing et al.    | no  | mmdetection2 | yes | 2x  | 40.8       | 37.4       | 40.1 (+2.7) |
> | Ours                          | yes | mmdetection2 | -   | 2x  | 41.0       | 37.4       | 40.6 (+3.2) |
> | Ours                          | yes | mmdetection2 | -   | 1x  | 41.0       | 36.4       | 40.1 (+3.7) |
> |||||||||
> | [14] Kang et al.       | no  | detectron2   | yes | 1x  | 40.4       | 37.4       | 40.7 (+3.3) |
> | Ours                          | yes | detectron2   | -   | 1x  | 40.4       | 37.4       | 40.7 (+3.3) |
> |||||||||
> | [5] Dai et al.        | no  | proprietary  | no  | 2x  | 38.1       | 36.2       | 39.1 (+2.9) |
> #### Table 2: Object detection accuracies by means of average precision (AP) using **RetinaNet** grouped by implementation framework. Reference results taken from the original papers. TS: Training Schedule, 1x = 12 epochs, 2x = 24 epochs, 2x* = 12 epochs using double batch size. C: whether code is publicly available? FB: whether the method is purely feature based?
>
> ### The explanation and theory analysis of the proposed method is limited.
> With this paper, the main goal is to provide intuition and motivation why higher order statistics like structure should be helpful cues in distillation methods just like in natural images. To provide some additional insight into why our method provides an effective measure of knowledge distillation, we re-state one of the main drawbacks of L2 (and L1): Their inability to take into account knowledge contained in spatial relationships between features. We solve this problem, by using additional local statistics between feature spaces (in our case cross-correlation and variance) as a distillation objective, which allows us to capture this additional knowledge. We show the effectiveness experimentally, and provide both qualitative and quantitative results. These results raise new questions to how we apply distillation in most SOTA works, which is why we want to make these insights available to the community.

---

> > ### Comment · Reviewer_6uNd · 2022-08-05
> > **Thanks for your response and additional results**
> >
> > The response of authors explan my questions and concerns, especially they add many results of comparison to more SOTA methods, so I decide to increase my score from 4 to 5.

---

> ### Author Response · Authors · 2022-08-02
> **Response to Reviewer 6uNd - Questions**
>
> this is a continuation of our previous response, in which we address the questions posed by the reviewer.
>
> ### Questions
>
> #### Q. In line 124, can you explame what normalization function you use to limit the value of feature maps to [0, 1]?
> A. For our experiments we used min-max normalization. In initial experiments of our research phase, we also investigated softmax normalization which slightly decreased performance (concretely a 0.8AP difference on the RetinaNetR101-R50 T-S pair in initial experiments). Therefore, the experiments in the submitted work all adopt min-max. We will include this additional piece of information in the camera ready version.
>
> #### Q. In the field of image classification, most KD methods based on feature can do a better performance with the combination of logits-based KD methods. I notice that your methods totally based on feature map, did or will you do some experiments on combining your distillation method with other distillation method?
> A. This question has been adressed in the general comment to all reviewers, as it showed similarity with a question posed by another reviewer.

---

### Author Response · Authors · 2022-08-02
**Addressing two similar lines of questioning by reviewers**

We thank the reviewers for their time and are pleased to hear that our work has been generally received favorably.
We would like to address two lines of questioning raised by multiple reviewers here, and respond to the individual questions in the respective answers.


## 1. Integration/Combination with other distillation methods.
### Questions
- "How will the performance change if replace original knowledge distillation objective in [14] with ssim?" **(XGaw)**
- "In the field of image classification, most KD methods based on feature can do a better performance with the combination of logits-based KD methods. I notice that your methods totally based on feature map, did or will you do some experiments on combining your distillation method with other distillation method?" **(6uNd)**

### Answer
We did not specifically combine our method with output distillation methods, we did however conduct experiments integrating our method with [33] and [14], which are both mechanisms that ultimately apply a (weighted) mask over the feature space. As can be observed in tables 1 and 2, both methods reduce performance compared to applying SSIM over the entire feature space. We hypothesize that the mask harms the ability of SSIM to focus on knowledge-dense regions, where no foreground object is located. We found these results to be less significant than others studied in the paper, and therefore due to space limitations we chose not to include these results in the main paper.




| Method             | $AP$                 | $AP_{S}$      | $AP_{M}$      | $AP_{L}$      |
|--|----------------------|---------------|---------------|---------------|
| Teacher            | 41.0                 | 24.1          | 45.3          | 53.8          |
| [33]        | 38.5 (+2.1)          | 21.7          | 42.6          | 51.5          |
| [33] + Ours | 39.0 (+2.6)          | 22.0          | 43.0          | 52.2          |
| Ours     | 40.1 (+3.7) | 23.1 | 44.6 | 53.2 |
#### Table 1: Combination of our methods with [33] where we replaced the MSE loss ($L_2$) after the spatial attention mask with SSIM.



| Method  | $AP$ | $AP_{S}$ | $AP_{M}$ | $AP_{L}$ |
|---------|------|----------|----------|----------|
| Teacher  | 40.4 | 24.1     | 45.3     | 53.8     |
| [14]        | 40.7 | 24.6     | 44.9     | 52.4     |
| [14] + Ours | 40.0 | 24.0     | 44.2     | 52.0     |
| Ours        | 40.7 | 24.0     | 45.0     | 53.1     |
#### Table 2: Combination of our methods with [14] where we replaced the MSE loss ($L_2$) with SSIM.


## 2. Application of our method in additional tasks
### Questions
- "If the distillation method doesn't depend on the heads, is it possible to extend it to a more general method that can be used for classification, instance segmentation (Mask R-CNN)?" **(cG4E)**
- "The SSIM loss works directly on feature maps, does not change object detection heads or losses. The methods proposed in the paper should works in all vision tasks such as classification, segmentation or recognition. However this is not mentioned in the paper." **(VXDw)**

We suggested to investigate the performance of our method in additional tasks as a potential for future work in our conclusion (265-266). State-of-the-art knowledge distillation methods for detection typically use feature-based guidance, where our method helps, while methods for semantic segmentation typically use a combination of feature and logit-based guidance. For object-centric tasks such as instance segmentation we would expect good performance improvement with our method, while in contrast on dense tasks such as semantic segmentation we expect limited benefit of naively applying SSIM.

---

### Meta-Review · Area_Chair_io6z · 2022-08-28

**Recommendation:** Accept
**Confidence:** Certain

**Metareview:**

The paper receives positive feedback after rebuttal. All reviewers agree that the idea of distilling structural knowledge for object detection is novel and worth sharing to the community. AC agrees with it and recommends accepting the paper.

**Award:**

No

---

### Decision · Program_Chairs · 2022-09-14

Accept